# Candida Genus Maximum Incidence in Boar Semen Even after Preservation, Is It Not a Risk for AI though?

**DOI:** 10.3390/molecules27217539

**Published:** 2022-11-03

**Authors:** Ștefan G. Ciornei, Dan Drugociu, Petru Roşca

**Affiliations:** 1Biotechnology of Reproduction, Department Clinics, Faculty of Veterinary Medicine, Iasi University of Life Sciences (IULS), M. Sadoveanu Alee, No 6, 700489 Iași, Romania; 2Reproduction and Obstetrics, Department Clinics, Faculty of Veterinary Medicine, Iasi University of Life Sciences (IULS), M. Sadoveanu Alee, No 6, 700489 Iași, Romania

**Keywords:** boar semen quality, microbiological profile, bacteryospermia, mycospermia, sow AI, sow fertility

## Abstract

There is little information in the literature about the fungal contamination of boar semen and its persistence during storage. The challenge of this study was to perform a mycological screening to identify the yeast in the raw semen at 12/24 h after dilution. The research was done in pig farms in the N-E area of Romania, with maximum biosecurity and state-of-the-art technology. All the examined ejaculates (101) were considered to be normal for each spermogram parameter, with microbiological determinations in T0 at the time of ejaculate collection, T1 at the time of dilution, and T2 at 24 h of storage. Microbiological determinations (mycological spermogram) were performed for quantitative (LogCFU/mL) and qualitative (typification of fungal genera) identification. Bacterial burden (×10^3^ LogCFU/mL) after dilution (T1) decreased drastically (*p* < 0.0001) compared to the one in the raw semen (T0). After 24 h of storage at 17 °C, the mean value of the bacteriospermia remained constant at an average value of 0.44. Mycospermia had a constant trend at T0 (raw) and T1 (0.149 vs. 0.140) and was slightly higher at T2 (0.236). The difference between T1 vs. T2 (*p* = 0.0419) was close to the statistical reference value (*p* = 0.05). Of the total genera identified (24), the fungi had a proportion of 37.4% (9/15) and a ratio of 1:1.6. Regarding the total species (34), the fungi had a frequency of 29.42% (10/24) with a ratio between the fungi and bacteria of 1:2.4. A fertility rate of 86% was observed in the L1 group (50 AI sows with doses and mycospermia from T1), and an 82% rate was observed in the L2 group (50 AI sows with doses and mycospermia from T2). The litter size of L1 was 9.63 piglets and 9.56 for L2. Regarding the total number of piglets obtained between the two groups, there was a slight decrease of 22 piglets in group L2, without statistical differences (*p* > 0.05). The predominant genera persisted after dilution during a 12 h storage at 17 °C, where yeasts, such as *Candida parapsilosis* and *C. sake* were identified in more than 92% of AI doses.

## 1. Introduction

The biosecurity of boar semen and the possibility of reducing contamination has a continuing interest in the pig industry. Some research is limited to identifying bacterial genera [1,2,3], others to calculating the degree of contamination and incidence [4,5] and others also show the harmful effect on sperm due to the decreasing viability [6] and reproduction indices in sows [7]. Current studies are developing new techniques to reduce contamination during sperm collection alongside new extenders that will replace and/or complement the effect of antibiotics [8]. Brescani C. et al. (2013) developed a new modified boar semen extender for short-term liquid storage based on the use of amikacin sulphate and fructose rather than gentamicin and glucose.

Similar to G. Althouse (2005), who introduced the term “bacteriospermia” for fresh and diluted semen, Ș. Ciornei (2021) introduced the term “mycospermia”, referring to the fungal contamination of boar semen [9]. Boar semen, unlike the sperm production of other males, is usually used diluted in liquid and stored at a relatively controlled temperature (between 14 and 18 °C). This technique requires the use of semen over a few days (5–7 days). Therefore, a special focus in the artificial insemination industry is to maintain the quality of doses that meet the fertilization objectives while maintaining bacteriological and virological safety [10].

A large number of studies have identified bacteria in boar semen, and these have been attributed to fecal, preputial, and cutaneous microorganisms or as originating from areas of semen doses processing for artificial insemination [3,11,12,13]. Martin M. et al. (2010) revealed that there is a positive correlation between the presence of *E. coli* and sperm agglutination.

In the pig industry, various procedures are used to reduce the microbial contamination of the sperm, such as automated semen collection technologies [14] or the use of antibiotics in extenders. Both of them lead to a bacteriological safety of the semen in doses for AI [15,16], but the fungal safety remains uncovered.

In addition to the goal of quality and safety (biosecurity), reducing a load of antibiotics in marketed semen is a goal of the pig semen industry [17].

Several bacteria in the Enterobacteriaceae family have been associated with significant decreases in sperm motility [18]. *E. coli* has been associated with a diminution which affects the litter size [7], while *P. aeruginosa* decreases sperm capacity in vitro [19].

Bacteriospermia is a major risk factor that should be avoided and controlled in order to prevent changes in sperm quality that could impair fertility performance in sows [1,11,20,21,22].

Seasonal variations are said to play important roles, affecting the quantity and quality of semen, which is of great economic importance to the pig breeding industry [23,24].

Some studies have shown that the season can affect a variety of quality characteristics in boar semen [25,26]. This highlights the quality parameters of semen that are affected by the season [27]. Zhang L. et al. (2022) revealed that boar semen microbiota in summer differed from that of winter semen, potentially due to seasonal changes related to semen quality and sperm fertilizing capacities. Higher bacterial diversity in the ejaculated semen was observed in winter rather than in summer [28].

There is little information in the literature about the fungal contamination of boar semen and its persistence during storage [29] or its effects on fertility [30].

Isolated publications have aimed at a complete asepticization of boar semen; therefore, Ciornei S. et al. (2021) identified the main elements of risk in bacterial and fungal contamination of semen and limited the degree of contamination by implementing hygiene protocols and the biosecurity of sperm collection (HPBC) [9].

There are regulations stating that antibiotics or a mixture of antibiotics should be added to extenders [31]. Therefore, all doses of semen intended for international trade should contain antibiotics, so they do not necessarily contain antifungals.

It would be logical to remove the bacteria from the semen sample rather than inhibiting their growth or killing them in doses. One such method for separating sperm from bacteria is colloidal centrifugation [31,32,33]. According to the model described above, it seems that it would work in the case of fungal contamination, but it must be studied further.

The addition of one or several antibiotics and antifungals to the sperm spreads reduces the bacterial and fungal load in the doses of semen for artificial insemination but can facilitate the development of antimicrobial resistance. It seems that this application is not therapeutic and does not match the current guidelines on the prudent use of these substances. Researchers have looked for alternatives to antibiotics, such as herbal extracts and nanoparticles, but they may not be effective in all situations or may be spermatotoxic, which is counterproductive. Colloidal centrifugation of the semen separates sperm from most bacteria, but it is not known exactly if this also applies to yeast. However, strict attention to hygiene should take placed at all stages of semen collection and processing [4,8,9].

The topic of the microbiological contamination of sperm and infertility is extremely interesting, and there are many aspects to cover. In particular, it is not known whether *Candida* genera affect sperm quality and reproductive potential.

This study aims to highlight the high degree of contamination in sperm with yeast, which have an endogenous origin and cannot be controlled by any method of biosecurity of sperm collection. More important is to know and demonstrate their effect in relation to the degree of contamination and the period of time.

## 2. Results

All the examined ejaculates were considered to be normal for each parameter of the spermogram. Each ejaculate corresponded to general and special assessments of the volume, odor, pH, sperm concentration, and motility. The samples had to accomplish the given criteria (volume > 200 mL, concentration > 0.2 × 10^9^ sperm/mL, motility > 70%) for subsequent procedures involving semen dilution and storage. Other organoleptic parameters used in the boar semen assessment were also normal.

The volume of the ejaculates had an average of 344.35 mL, the average concentration was 0.36 × 10^9^ sperm/mL, the total motility was 80% (with limits between 78 and 82), and the progression of the sperm was 39% (with limits of the 36 the 42). The quality of the analyzed semen was adequate. At 24 h, at the time of T2, the average motility was 69.5%, and the progressiveness was 31.5 (with limits from 27 to 36%).

### 2.1. Microbiologial Profile of Semen

Burden LogCFU/mL

Following microbiological analysis to identify the number of germs per milliliter of semen, both bacteria and fungi were identified. The data are presented in Table 1 as averages of the three microbial load control moments T0 (ejaculate collection time), T1 (after dilution and dose preparation), and T2 (24 h after storage). Additionally presented are the statistical data with the *p*-values.

The bacterial load in the raw semen (at time T0) had an average concentration of 82.41 × 10^3^ LogCFU/mL with quite wide variations depending on the sanitary rigor of the collection. After using antibiotics, the average number of bacterial colonies decreased drastically and were statistically significant (*p* < 0.0001) at the time of T1 (control) with an average of 0.354 × 10^3^ LogCFU/mL. After 24 h of dilution and storage at 17 °C, the mean value of the bacteriospermia remained constant at an average value of 0.449 × 10^3^ LogCFU/mL (Table 1).

The fungal load of the raw sperm registered values of T0 of 0.149 × 10^3^ LogCFU/mL, at the moment T1 of 0.140 × 10^3^ LogCFU/mL and at the moment T2 of 0.236 × 10^3^ LogCFU/mL (Table 1).

The total number of germs, including bacteriospermia and mycospermia was 82,559 × 10^3^ LogCFU/mL at the time of collection, 0.494 × 10^3^ LogCFU/mL at 12 h, and 0.685 × 10^3^ LogCFU/mL at 24 h.

### 2.2. Genus/Species and Frequency

The qualitative examination of the sperm consisted of the identification of the genera and species of the bacteria and fungi, followed by the establishment of the frequency in their isolation. The determinations were performed at two key points: at the time of ejaculate harvesting (T0) and 24 h after dilution and strain (T2).

The results described in Table 2 show that 24 genera and 34 species of bacteria and fungi were identified. Of the total sperm microbiota genera identified (24), fungi accounted for 37.4% (9/15), and the ratio was 1:1.6 between the fungal and bacterial genera. Regarding fungal species (34), they had a weight of 29.42% (10/24) with a ratio of fungi to bacteria of 1:2.4.

The frequency of the bacterial isolation in freshly collected ejaculates (T0) was as follows in descending order: *E. coli* 81.2%, *Staphylococcus* 72.3%, *Pseudomonas* 63.4%, *Streptococcus*, and *Enterococcus* 45.5%, *Proteus* 35.6%, *Yersinia*, *Tatumella*, *Pantoea*, *Serratia* and *Shiqella* 26.7%, *Actinomyces*, *Bacillus*, and *Arcanobacterium* 10.9%, and *Klebsiella* 6.9% (Figure 1 and Figure 2).

After diluting the semen and storing it for 24 h, the frequency of the isolation decreased, most reaching the minimum value (Figure 2). Several genera showed isolation frequencies between 3.5% (*Escherichia*, *Enterococcus*) and 9.8% (*Arcanobacterium*).

Regarding the evolution of the fungal isolation from semen (T0), it was found that of the 10 isolated genera, the highest frequency was *Candida* (92.1%) and *Geotrichum* (72.3%), followed by *Aspergillius* and *Penicillium* (63.3%), *Mucor* (45.5%), *Cladosporium* and *Fusarium* (36.6%), and *Alternaria* and *Acremonium* (18.8%) (Figure 3).

After its dilution and preservation at 17 °C, the frequency of the fungal isolation followed a constant trend, reaching values similar to those of the fresh semen (Figure 4 and Figure 5). It should be noted that the *Candida* genus (*parapsilosis*, *sake*) identified had an identification frequency of 94.1% (T2).

### 2.3. Fertility of Sows with Doses of Semen with Known Mycospermia

After preparing the doses and the insemination of the programmed sows, the results are presented in Table 3. Fertility of 86% was observed in group L1 and 82% in group L2. Lot L2 came from AI with semen preserved for 24 h and with a mycospermia of 0.236 × 10^3^ LogCFU/mL.

Litter size (prolificity) had an average of 9.63 piglets in one farrowing for group L1 and 9.56 in group L2. Regarding the total number of piglets obtained between the two groups, there was a slight decrease of 22 piglets in group L2. However, no significant decreases were found between L2 and L1. There were no statistical differences (*p* > 0.05) between fecundity and prolificity.

## 3. Discussion

The current study, which only aimed to signal the *Candida* spp. genera in preserved boar semen, we used only the classic and Miniapi methods. According to studies in the literature [34], the Conventional Identification (CI)-MiniApi method compared to MALDI-TOF [35,36] or MT-MS is slower and less faithful but generates the same results. According to Sendid B. et al. (2013), concordance between the two techniques is excellent for the medically important species (98–100%), including the identification of closely-related species (*Candida* genus).

After qualitative microbiological examinations, it was reconfirmed that boar semen had a variable degree of bacterial and fungal contamination. Numerous studies have highlighted the bacterial contamination [1,2,3,4,5,6,7,9,37] of boar semen and its negative effects, both in terms of the number of germs (LogCFU) and their typing [11,13,20]. In general, when talking about the contamination of boar semen and its quality in preservation, attention is directed toward the bacterial one. However, it seems that the microbiology of sperm (especially boar) has much more complex valences because, after collection, the semen can contain all kinds of microbes. In terms of the influence on the quality and parameters of reproduction, bacteria and fungi are involved. In terms of the spermicidal effect, sperm viruses have little or no influence on the quality of semen for AI. Indeed, sperm may contain various viruses [10], but they serve as a route of transmission and not as a direct aggression of sperm or of the dilution and preservation environment, which makes it not an objective of this study.

Researchers that have induced the pig industry development revealed that sperm collection in pigs is not and should not be a sterile procedure, but to ensure that the microbiological biosecurity of AI and the continuous optimization of technologies are needed. Techniques (hygiene) and methods (extenders) that reduce or annihilate the harmful effect are currently used in pig breeding centers [1,5,7,28,37].

The success of natural fertility is the result of the ability of sperm to fertilize oocytes and the biosecurity of the uterine environment. In the mechanism of animal reproduction, microorganisms are undoubtedly at stake [3,13,20]. The relationship is essential: the degree of contamination in the sperm and the natural immunity of the seminal plasma and the uterus. When these balances are broken by a microbial load, fertility suffers. In assisted reproduction, they attempt to control through sanitary techniques of collecting and manipulating sperm and by using antimicrobial extensions, but sometimes the methods become ineffective (with reference to mycospermia) [20,38].

Semen contamination was detectable in more than 90% of the samples, with only 8% of samples exhibiting values less than 10 LogCFU/mL. Some authors reported similar or higher percentages in their papers [3,11,13,39].

In this study, after microbiological determinations were performed on the boar semen, it was observed that this raw collection has a variable burden on the total microbial contamination with 82.559 × 10^3^ LogCFU/mL of bacteria of 82.4 and fungal of 0.149 × 10^3^ LogCFU/mL. In these conditions, the average sperm parameters were 80% motility (M%) and 69.5% progressivity (P%). After contact with antibiotics in the extenders, the microbiological cargo suffered a decrease to 0.494 × 10^3^ LogCFU/mL, an anticipated and statistically significant action (*p* < 0.0001).

This significant decrease was caused by the reduction in the bacteria observed at 12 h after dilution (T1 time compared to T0) with an average of 0.354 × 10^3^ LogCFU/mL. While fungi in semen diluted at 12 h had, on average, almost the same values as at T0, 0.140 × 10^3^ LogCFU/mL, which showed that the diluents had no role in reducing mycospermia.

While at the time of T2 and 24 h of storage at 17 °C, the total microbiological cargo did not undergo visible changes from 0.494 to 0.685 × 10^3^ LogCFU/mL (statistically not significant *p* = 0.08), this was almost the same as the bacterial evolution (0.449 × 10^3^ LogCFU/mL), with 39% motility (M%) and 31.5% progressivity (P%). Attention was focused on the evolution of the fungal load, which also made the difference, so there was a small increase from 0.236 to 0140 × 10^3^ LogCFU/mL. The difference is statistically representative (*p* = 0.0419) but close to the statistical reference value (*p* = 0.05).

This increase in the number of fungi during the preservation seems to be argued by the inefficiency of the extensions, the yeasts develop at 17 °C, and all the fungi consumed glucose and fructose.

There are some natural mechanisms for reducing the microbiota at the time of natural mounting in the relationship between ejaculation and fertility. Several animal species have been known to possess antibacterial defenses in their ejaculation to maintain fertilization and the biosecurity of the genital tract. Studies on the immunity associated with boar and bull ejaculation are known to show bacterial destruction factors and lysosomal enzymes [40,41]. The results indicate that *Lactobacillus* is not only a probiotic potential for sperm quality and fertility potential but may also be beneficial in limiting the negative influence of *Pseudomonas* [28,42].

Uterine immunity during estrus is a known mechanism that prevents possible microbial invasions, but the result is uncontrollable depending on the degree of contamination and the ability of the individual female [43,44].

The use of antibiotics in extenders, so far, is the most used method to reduce bacterial contamination [45,46], but with some negative effects (such as inefficiency on fungi and the creation of antimicrobial resistance). Colloidal centrifugation and other techniques/methods could optimize procedures that are more difficult to apply in the industry [4,9,34,45,47,48].

It remains that the most important measure of control in the boar sperm microbiota is the hygiene of the sperm collection and handling protocol. According to the previously validated measures Ciornei S. et al. (2021), following an HPLC protocol, demonstrated a reduction in sperm contamination with germs from both external and internal sources. Therefore, the bacteria and filamentous fungi would be under control, remaining that only the yeasts of the genus *Candida* persisted in the conservation doses. They seemed to have an endogenous origin which, in this study, was represented at 0.236 × 10^3^ LogCFU/mL.

From the point of view of the typification of the microbial genera and species, our study revealed the same genera and species as in other authors and that the frequency of their identification varies from one author to another [1,2,3,4,5,6,7,11,37].

Some similar publications reported higher frequencies in bacterial identification, such as that of Costinar Luminita et al. (2022), instead of other research, such as Martin et al. (2010), who reported results more similar to ours [37,49].

Of all the germs we identified through this study, the *Candida* genus had the highest incidence in identification. Thus, at time T0, it had a 92.1% and at T2, a 94.2% increase in the neutral area of statistical correlation (*p* = 0.051). Regarding the standardized species, *C. sake* and *C. parapsilosis* were identified with a constant frequency, even after dilution in the storage medium.

Yeasts of the genus *Candida* have an endogenous origin in the segments of the genital system of the boar, and they do not seem to be influenced too much (such as filamentous fungi and bacteria) by the sanitary methods of sperm collection nor by the component of diluents. Moreover, continuous *Candida* mycospermia develops in doses intended for the preservation and insemination of sows. Therefore, this parameter should be followed in terms of the quality of the boar semen used in AI.

After our study on the insemination of groups of sows with known mycospermia, some small but statistically insignificant differences were obtained (*p* = 0.279).

Any influence on fertility and prolificity was reflected in the number of piglets for fattening and slaughter. From this point of view, pig farms with thousands of farrowing per year and better management of these phenomena can bring economic profit. Therefore, the prolificity (+0.02), fecundity (+4%), and the total number of piglets (+22) were better in the group with a lower mycospermia (0.141 × 10^3^ LogCFU/mL), and so there are perspectives and new research interests in order to improve the processes of assisted reproduction in pigs.

At present, many details are known regarding the microbiota of boar semen [42,48,49], with implication effects and measures, but they are considered without too much influence by fungal contamination. A load of sperm with fungi can influence the reproduction processes in the pig industry and deserves attention in discovering new knowledge in this field [1,4,5,6,7,12,15,28,34,37].

Through all these recommendations, yeasts of the genus *Candida* spp. exactly cannot be controlled because they are of endogenous origin. It would be encouraging that for a short period of time (12 h) from the preparation of the doses no harmful and counterproductive effects would appear. It remains to be seen, in the future, through more research and studies.

The aim of this study is to highlight a less studied and known aspect related to the boar sperm microbiota, with direct reference to the increasing persistence in the conservation and AI doses of yeasts of the genus *Candida*.

## 4. Materials and Methods

### 4.1. Study Design and Animal Sampling

All procedures involving animals were carried out in accordance with guidelines and regulations according to the European Commission Directive for Pig Welfare. All experimental boars were exposed to the same rearing conditions. Experiments complied with the standards of institutional guidelines for ethics in animal experimentation. All experimental procedures were permitted by the Animal Ethics Committee of IULS-FVM and farmers (EU 2010/63 and National directives Ord. 28/31–08–2011 and National Law 206/2004).

### 4.2. Pig Farms

The pig farm where the research took place is located in the North-East of Romania. This farm was chosen because it is one of the largest and most modern farms in Romania and has a large flow of animals where the conditions of biosecurity, shelter, fodder, and welfare are strictly observed. Before, during, and after our research on the farm, African Swine Fever did not evolve. The endowments of the farm and of the reproduction sector and AI are at State of the art levels.

### 4.3. Biological Material

The genetic material used for breeding were boars and sows of a high genetic value (PIC, Petrain, Durok), which cross-produced and delivered line piglets from fatteners. During the research, the animals were clinically healthy, with a normal diet of feeding and deworming. The maintenance and exploitation of the animals were optimal, in accordance with European standards.

### 4.4. Reproduction Organization and Study Design

In the Reproduction Laboratory of the farm, AI biotechnologies are planned daily, for which the sows are detected in estrus, and the optimal moment is calculated. Boars used for sperm production are staggered weekly for collection according to a well-planned program [50,51,52]. Each ejaculate is examined, and doses of AI are prepared immediately afterward (3–5 days). The processes of the collection, examination, dilution, preservation, and insemination of semen were similar.

### 4.5. Collection of Semen and Study Samples

The boars were prepared for prey-semen collection by emptying the prepuce diverticulum, washing, and disinfecting the region. The sperm collection room was equipped with sow mannequins which were sanitized according to their own biosecurity protocol: a mist spray-fog of lightly decontaminating substances (Misoseptol) was introduced 15 min before the start of the collection. The operator equipped with two vinyl gloves cleaned and sanitized the prepuce region, then the first glove was removed, and the collection was started with the remaining glove. Sperm-rich fractions were collected by a qualified technician using the gloved-hand technique and were subsequently immediately transported to the laboratory in an isothermal vessel. Vessels for the working semen, including glassware, plasticware, and containers, were sterilized. Under sterile conditions, 10 mL of each ejaculate was collected in tubes to determine the microbiological spermogram.

### 4.6. Examination of Ejaculate Quality, Dilution and Dose Preparation for AI

The evaluation of the quality of each ejaculate was performed according to the methodology described in specialized practical guides [50,52] and grouped into two categories: general and special examinations [53].

The color was assessed visually by observing the semen in transparent microtubes, examining the degree of turbidity, the presence of blood, or an unusual color. The volume was assessed by direct observation in the graduated sampling container. Sperm concentration was determined immediately after ejaculate collection using the AccuRead Sperm Counter spectrophotometer (IMV-Technologies, L’Aigle, France).

Motility (M%, total sperm motility; P%, total sperm progressivity) was evaluated by the CASA system (computer-assisted sperm analysis) [38], produced by Hamilton—Thorne Bioscience, using the Animal Motility Software, Viadent option. For the sample analysis, Leja blades of 30 µL, with 4 chambers (Cryo BioSystem, France), were used in which 10 µL of semen were placed [52].

The corresponding ejaculates were diluted with a commercial extender (TRIXcell+) that ensured long-term preservation (5–7 days) and contained a combination of antibiotics [42]. For each insemination dose, 3.5 billion sperm were assigned in a volume of 80 mL.

From the diluted semen, 10 mL were collected in sterile tubes and deposited for 24 h at the same storage conditions as the AI doses.

### 4.7. Microbiological Spermogram (Bacteriospermia and Mycospermia)

To determine the bacteria and fungi in the sperm, one hundred and one samples were collected from each ejaculate, and fifty-one samples after dilution at 12 and 24 h.

### 4.8. Quantitative Determinations (LogCFU)

The total number of viable bacteria and fungi was assessed using the serial dilution method and incubation in aerobic conditions. A series of ten-fold dilutions (10−1, 10−2, 10−3) of the semen samples was performed using tubes containing phosphate-buffered saline (Biokar, Allonne, Oise, France). From each dilution, six volumes of 100 μL were plated on six Petri dishes containing solid media. Three of them contained tryptone soy agar (Biokar, France) and were used for the enumeration of bacteria after an incubation of 24 h at 37 °C, and the others containing Sabouraud Chloramphenicol Agar (Biokar, France) were used for the enumeration of fungi after an incubation of 3 days at 25 °C. Finally, the bacterial and fungal burden in the raw and diluted semen was calculated and expressed as LogCFU/mL, according to methods by APHA [54]. The total viable aerobic count per mL consisted of a LogCFU number of bacteria and a LogCFU number of fungi.

The number of germs was calculated at three different times: T0- at the time of ejaculate collection (from the raw semen), and T1- at the time of dilution and preparation of the AI doses, to control the bactericidal effect of the diluent and T2—at 24 h of storage.

### 4.9. Qualitative Determinations (Identification of Fungal and Bacterial Genera)

The determinations were performed on semen samples collected at T0 and T2. To identify bacterial contamination, each sample was placed on the plate onto the Columbia agar with 5% (*v*/*v*) sheep blood and Mac-Conkey agar, respectively. The plates were incubated in aerobic conditions at 37 °C for 24 h. After this period, one colony of each morphotype was transferred onto the tryptone soy agar and re-incubated for 24 h at 37 °C in order to obtain a fresh culture ready for identification. The identification of the bacteria isolates was performed using Gram stain and ID32E, ID32GN, ID32STAPH, and ID32STREP (bioMérieux, Craponne, France). Filamentous levura (fungi/yeasts) were identified on the basis of macroscopic and microscopic features using the primary cultures onto a Sabouraud Chloramphenicol Agar. The yeasts were identified by biochemical tests using ID32C strips (bioMérieux, France) [22,54].

#### 4.9.1. Observations on the Effect of Mycospermia on Sow Fertilization

Through this working protocol, we tried to find possible influences of the dose load with *Candida* spp.

Two homogeneous groups of primiparous sows were organized at the first voluntary estrus after weaning. The sows were carefully monitored to manage the preovulatory moment. Single insemination was performed by the Golden Pig catheter endocervical method (IMV, France). The diagnosis of pregnancy was made by ultrasound. Breeding indicators with interest in the farm, including fertility, farrowing rate, and prolificacy were calculated.

Group 1 (L1), consisting of 50 sows, was AI with doses of semen immediately after dilution and dose organization (T1).

Group 2 (L2), consisting of another 50 sows, was AI with doses of preserved semen after 24 h (T2).

Wanting to compare the AI results with the doses of semen with known microbiology (constant and persistent fungal contamination), this protocol followed the effect on the fertility of sows with known mycospermia at the time of preparation (T0) and after 24 h of storage (T2).

Both groups had the same conditions on the farm in terms of shelter—nutrition—comfort—and well-being management.

#### 4.9.2. Statistical Analyses

Basic descriptive statistics of the one-way ANOVA test and t-Test were used to interpret the obtained data. For the ANOVA test, *p* < 0.05 was considered statistically significant. Statistical analysis was performed with Prism version 8 (GraphPad Software 5.0. La Jolla, CA, USA, www.graphpad.com, accessed on 7 July 2022.).

## 5. Conclusions

Following this study, it was observed that bacteria and filamentous fungi could be controlled by sanitary procedures and by using commercial extenders. At 12 and 24 h after dilution and preservation at 17 °C, only the load in yeast fungi persisted and even increased. *Candida parapsilosis* and *C. sake* were identified in more than 92% of the AI doses. The fertility of the AI sows with a lower mycospermia generated better results compared to the high one. Therefore, there are perspectives and new research interests in order to improve the processes of assisted reproduction in pigs.

## Figures and Tables

**Figure 1 molecules-27-07539-f001:**
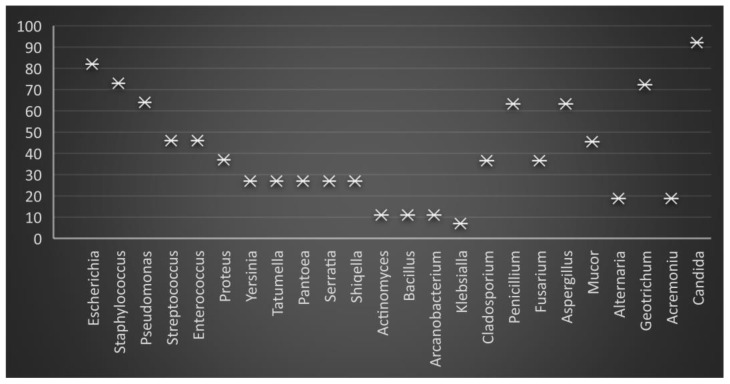
Graphic presentation of the percentage of microbiological load in freshly collected (raw) boar semen.

**Figure 2 molecules-27-07539-f002:**
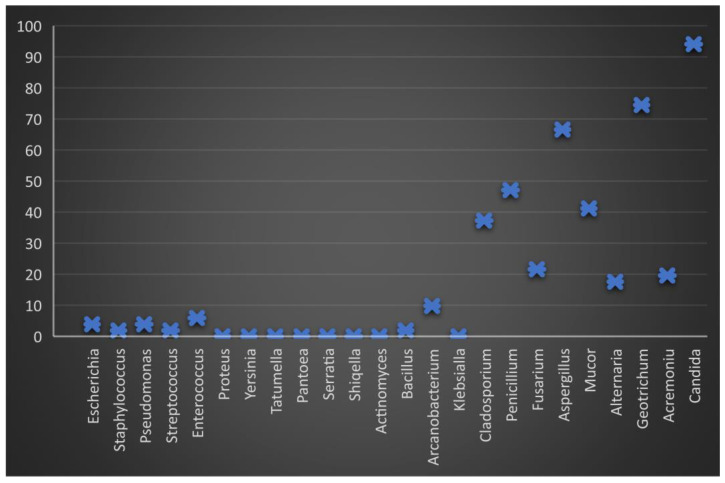
Graphic representation of the percentage of microbiological load in boar semen 24 h after dilution and storage at 17 °C. The minimum level of bacterial load is observed, but the fungal load remains constant, especially the genus *Candida*.

**Figure 3 molecules-27-07539-f003:**
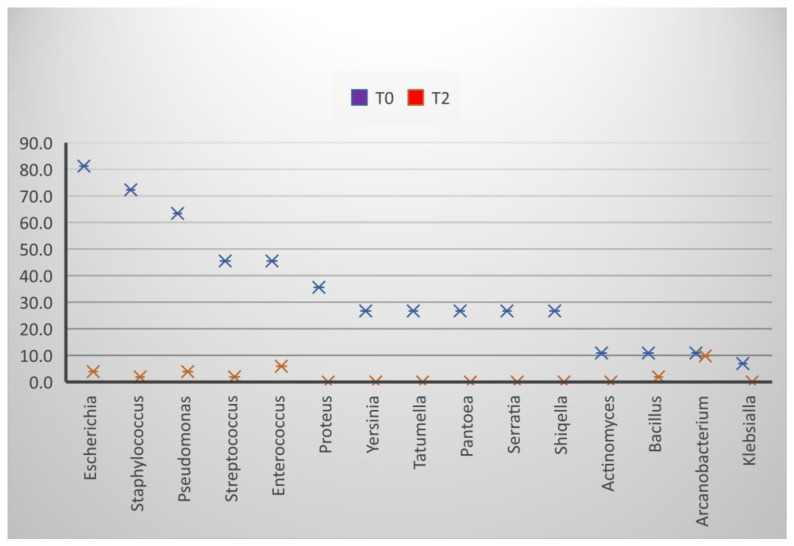
Effect of extenders on the frequency of bacterial isolation, T0 at the time of collection, T2 24 h after dilution.

**Figure 4 molecules-27-07539-f004:**
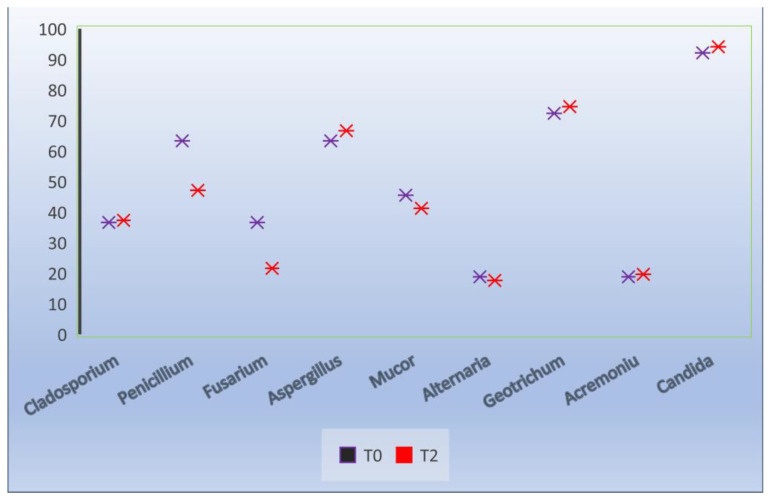
Effect of extenders on the frequency of fungal isolation, T0 at the time of collection, T2 24 h after dilution.

**Figure 5 molecules-27-07539-f005:**
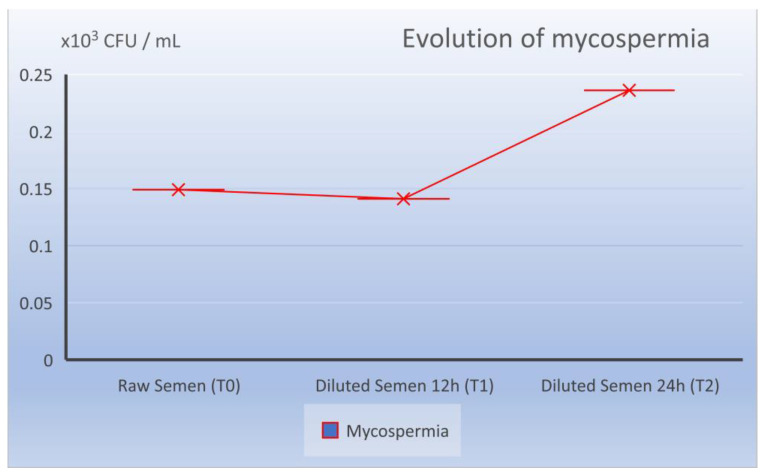
Evolution of mycological load in the dynamics from ejaculate collection (T0), 12 h after dilution (T1), and 24 h preservation (T2).

**Table 1 molecules-27-07539-t001:** Microbiological load (bacterial and fungal) of boar semen at the time of ejaculate collection (raw semen T0), 12 h after dilution (T1) and 24 h after dilution and storage at 17 °C.

10^3^ LogCFU/mL	Bacteriospermia	Mycospermia	Total Germs
Raw Semen (T0)	82.41	0.149	82,559
Diluted Semen 12 h (T1)	0.354	0.141	0.495
Diluted Semen 24 h (T2)	0.449	0.236	0.685
Statistical Significance (p)	T0 vs. T1 <0.0001 (yes)	T0 vs. T1 = 0.1728 (no) T1 vs. T2 = 0.0419 (yes)	T0 vs. T1 <0.0001 (yes) T1 vs. T2 = 0.081 (no)

**Table 2 molecules-27-07539-t002:** Microbiological profile of raw boar semen. Identified bacterial and fungal genus and species: mycospermia (A) and bacteriospermia (B).

	Genus	Species	Frequency of Isolations % (*n*)	
			Raw Semen (T0) *n* = 101	Diluted Semen (T2) *n* = 51
Fungus	Cladosporium	cladosporoides	36.6 (37)	37.3 (19)
	Penicillium	spp.	63.3 (64)	47.1 (24)
	Fusarium	spp.	36.6 (37)	21.6 (11)
	Aspergillus	spp.	63.3 (64)	66.6 (34)
	Mucor	racemosus	45.5 (46)	41.2 (21)
	Alternaria	alternata	18.8 (19)	17.6 (9)
	Geotrichum	candidum	72.3 (73)	74.5 (38)
	Acremoniu	spp.	18.8 (19)	19.6 (10)
	Candida	parapsilosis, sake	92.1 (93)	94.1 (48)
B. Bacteria	Escherichia	coli	81.2 (82)	3.9 (2)
	Staphylococcus	aureus, zooepidemicus, intermedius, hiyicus	72.3 (73)	1.9 (1)
	Pseudomonas	aeruginosa	63.4 (64)	3.9 (2)
	Streptococcus	suis	45.5 (46)	1.9 (1)
	Enterococcus	faecium, faecalis	45.5 (46)	5.9 (3)
	Proteus	vulgaris	35.6 (37)	0 (0)
	Yersinia	enterocolitica, ruckeri, pseudotuberculosis	26.7 (27)	0 (0)
	Tatumella	ptyseos	26.7 (27)	0 (0)
	Pantoea	spp.	26.7 (27)	0 (0)
	Serratia	ficaria, marcescens	26.7 (27)	0 (0)
	Shiqella	spp.	26.7 (27)	0 (0)
	Actinomyces	suis	10.9 (11)	0(0)
	Bacillus	subtilis, cereus, megaterium	10.9 (11)	1.9 (1)
	Arcanobacterium	pyogenes	10.9 (11)	9.8 (5)
	Klebsialla	pneumoniae	6.9 (7)	0 (0)

**Table 3 molecules-27-07539-t003:** Fertility and comparative prolificacy of sows inseminated with doses of semen with known mycological load.

Sow AI (n)		Doses		Fecudity (%)	Prolificity	
		Sperm × 10^9^	Mycospermia LogCFU/mL × 10^3^		Average	Total Piglets
L1	50	3.5	0.140 (T1)	86 (43/50)	9.63	414
L2	50	3.5	0.236 (T2)	82 (41/50)	9.56 *	392

* Statistical significance *p* = 0.279; There were no statistical differences (*p* > 0.05) between the fecundity and prolificity occurring in the control and experimental groups.

## Data Availability

The raw data supporting the conclusions of this article will be made available by the authors, without undue reservation.

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
