# Peer review of "Candida Genus Maximum Incidence in Boar Semen Even after Preservation, Is It Not a Risk for AI though?"

_molecules, 2022, doi:10.3390/molecules27217539_

Round 1
Reviewer 1 Report
Please see attached file for comments

Author Response
Thank you for your critical comments on this manuscript, this is of course for the purpose of science and innovation
All your suggestions are welcome, they have been processed and I hope it will improve the quality of our manuscript.
The purpose of this study was to highlight the real contamination of sperm in fungi, especially yeast-like ones. From this step promoters, there are perspectives and new research interests in order to improve the processes of assisted reproduction in pigs.
To reach this point, we evaluated the boar sperm microbiota at collection, dilution and after. It was necessary to highlight the fact that through specific collection biosecurity techniques and through the antibiotics used in diluents, bacteria and filamentous fungi can be counterbalanced. By comparison, the yeast fungi seem to remain in the sperm and even multiply during preservation.
I have made the corrections you recommended to the text.
Regards.

Reviewer 2 Report
Abstract: see different size of font on the beginning of abstract and the othe parts – uniform it
Introduction: Introduction seems too long, please shorten it if its possible
Material and methods:
1. Please can you note which kind of extender with what antibiotics were used in this study?
2. Why you used only Columbia and MacConkey agar? Semen can contain also other specified bacterial strains expect these which you detected?
3. Also why you did notu sed anaerobic condition for bacterial cultivation. Semen is a sample with limited content of oxygen. Therefore it can contain some anarobic species.
4. logCFU should be more representative as CFU
Results:
1. if bacterial count dramaticaly decrease after antibitoics addition, then antimicrobial resistance of cultivated bacterial strain should be used. Minimal inhibition concentration should be avaluated and after evaluation increase antibiotic level for inhibition of all groups of bacteria.
2. Graphical presentation of bacterial and fungal load should be better and clearer presented – try to prepare graph by Krona: link to github - https://github.com/marbl/Krona/wiki
3. Results show that extender do not contain antimycoticum, only antibacterial antibiotics, therefore its important to inform about contents of extender
4. Article with the similar problematic, can be cite: https://www.mdpi.com/1420-3049/24/23/4329
Discussion: ca be improved
Conclusion: Article has a very simply methodology, but the scienfitic contributiuon is considerable, because topic of this study is important for prolong viability of the sperm for pig reproduction, also for another animal reproduction sphere. I mean that article can be published after minor revision which means previously.
Thank you
Author Response
Thank you for your competent comments, I hope that your suggestions have raised the level of the article and that it will bring new contributions in the field of assisted reproduction in pigs. An important sector of the meat industry today and in the future.
Regards.

Reviewer 3 Report
This article deals to highlight the high degree of contamination of sperm with yeast, which have an endogenous origin and cannot be controlled by any method of biosecurity of sperm collection. More important is to know and demonstrate their effect, in relation to the degree of contamination and the period of time.
In my opinion, the article deserves publication in Molecules, but the molecular identification of isolates section should be improved according to the following items.
1) Article not contain novelty results.
2) Microbiological Analysis is very simple and these isolates need to be confirmed by molecular identification such as 18 rDNA. Also need to add a phylogenetic tree for isolated Candida.
3) Abstract need to rewrite and references are not in the format of the journal.
4) Unfortunately, the microbiological studies shown by the authors are not optimal and this is not acceptable from a quantitative point of view.
5) All microorganisms' names must be checked and changed to italic.
6) There are a lot of punctuation and typographical errors throughout the manuscript. Unfortunately, I can’t correct it throughout. It must be rechecked by a native English speaker.
7) Most of the authors of the references cited are too old and few references from the last 5 years. The author should look at recent advancements in literature and recommend citing a few of them.
Minor items are:
- The authors sometimes use abbreviations and sometimes use other abbreviations. Please check and correct.
- The English should be checked.
Author Response
Thank you for spending time to evaluate our article entitled: Candida Genus maximum incidence in boar semen even after preservation, isn't it a risk though for AI?
All your comments and suggestions are welcome and will improve the quality of our manuscript. We made in the text the corrections you recommended. Likewise, I have included below some answers and explanations to your suggestions.
Regards

Round 2
Reviewer 1 Report
Thank you for your responses to my concerns. I still maintain that the differences you see in the fertility of the 2 cohorts could well be due to differing storage times and not fungal load. The experimental design needs to be altered to be able to control for this. It is for this reason that I believe that this article is not worthy of publication, unless this material is removed.
Reviewer 3 Report
The article improved